# Longitudinal Assessment of Serum 25-Hydroxyvitamin D Levels during Pregnancy and Postpartum—Are the Current Recommendations for Supplementation Sufficient?

**DOI:** 10.3390/nu15020339

**Published:** 2023-01-10

**Authors:** Pilar Palmrich, Alexandra Thajer, Nawa Schirwani, Christina Haberl, Harald Zeisler, Robin Ristl, Julia Binder

**Affiliations:** 1Department of Obstetrics and Feto-Maternal Medicine, Medical University of Vienna, 1090 Vienna, Austria; 2Division of Neonatology, Department of Pediatrics and Adolescent Medicine, Pediatric Intensive Care Medicine and Neuropediatrics, Comprehensive Center for Pediatrics, Medical University of Vienna, 1090 Vienna, Austria; 3Center for Medical Statistics, Informatics and Intelligent Systems, Medical University of Vienna, 1090 Vienna, Austria

**Keywords:** maternal medicine, pregnancy, vitamin D, hypovitaminosis

## Abstract

(1) Background: Pregnant women are at risk of vitamin D deficiency. Data on pregnancy outcomes in women with vitamin D deficiency during pregnancy are controversial, and prospective longitudinal data on vitamin D deficiency with consistent definitions in pregnant women are scarce. (2) Methods: The aim of this prospective longitudinal cohort study was to investigate 25-hydroxyvitamin D levels over the course of pregnancy and postpartum in singleton and twin pregnancies with regard to dietary and supplemental vitamin D intake and environmental factors influencing vitamin D levels, evaluated by a standardized food frequency questionnaire. (3) Results: We included 198 healthy singleton and 51 twin pregnancies for analysis. A total of 967 study visits were performed over a 3-year period. Overall, 59.5% of pregnant women were classified as vitamin D deficient in the first trimester, 54.8% in the second trimester, 58.5% in the third trimester, 66.9% at birth, and 60% 12 weeks postpartum, even though 66.4% of the study population reported daily pregnancy vitamin intake containing vitamin D. Dietary vitamin D intake did not affect vitamin D levels significantly. (4) Conclusions: The majority of pregnant women evaluated in this study were vitamin D deficient, despite administration of pregnancy vitamins containing vitamin D. Individualized vitamin D assessment during pregnancy should be considered to ensure adequate supplementation and prevention of hypovitaminosis D.

## 1. Introduction

Vitamin D deficiency has been acknowledged as a worldwide health issue. Pregnant women have been identified as a population at increased risk of vitamin D deficiency, potentially resulting in adverse pregnancy outcomes [1,2,3]. The definition of a healthy vitamin D status during pregnancy, measured by circulating serum concentrations of 25-hydroxyvitamin D (25(OH)D), has been subject of debate, the most frequent definition being 25(OH)D levels < 75 nmol/L (30 ng/mL) [4]. However, a definition consensus is lacking. The prevalence of vitamin D deficiency during pregnancy ranges from 18% to 84%, also depending on the country and altitude of residence, skin pigmentation, and local clothing customs [2,5,6,7]. During pregnancy, vitamin D metabolism shows notable differences compared to the non-pregnant state. Studies show that maternal 25(OH)D crosses the placental barrier and constitutes the main source of vitamin D to the fetus [8], and the expression of the vitamin D receptor and regulatory metabolic enzymes has been found in the placenta and decidua, highlighting a potential pathway in the immunomodulation at the maternal–fetal interface [8]. Considering these effects and the high prevalence of vitamin D deficiency, the impact of vitamin D deficiency during pregnancy is a topic of much interest. The well-known effects of vitamin D deficiency during pregnancy and in neonates are late hypocalcemia and nutritional rickets [3,9]. Vitamin D has been shown to boost innate immunity [10], and various studies have found that the prenatal vitamin D status has an effect on the child’s susceptibility to develop asthma in later life [3,11]. There have been a number of observational studies demonstrating a link to relevant complications of pregnancy, such as hypertensive disorders of pregnancy, gestational diabetes, fetal growth restriction, and preterm birth [2,8,12,13]. However, to date, only a few large randomized controlled trials have been conducted and have generated conflicting evidence for the role of vitamin D supplementation in improving perinatal outcomes [14]. Existing data lack comparability due to varying definitions of vitamin D deficiency and inconsistent correction for influencing factors, such as the BMI (body mass index), sunlight exposure, and dietary intake. Due to a lack of evidence for nutritional requirements of 25(OH)D, current dietary recommendations vary between 5 and 15 micrograms/day (200–600 IU), with serum 25(OH)D targets of >50–75 nmol/L (20–30 ng/mL) [4,15,16]. The current recommendations followed in Austria vary between 200 and 800 IU per day [17,18]. The evidence to define clinical recommendations regarding vitamin D supplementation during pregnancy is currently insufficient, and prospective longitudinal data on vitamin D deficiency in pregnant women are missing. We therefore aimed to evaluate maternal 25-(OH)D levels and the effects of dietary and supplementary vitamin D intake over the course of pregnancy. Furthermore, we investigated a possible association of vitamin D deficiency with maternal and perinatal complications.

## 2. Materials and Methods

In this prospective observational study at the Department of Obstetrics and Fetomaternal Medicine at the Medical University of Vienna, we evaluated 25-hydroxyvitamin D levels over the course of pregnancy and postpartum in 199 low-risk singleton and 52 twin pregnancies between April 2012 and February 2016. The study was approved by the local research ethics committee (approval number 1051/2011) of the Medical University of Vienna. Written informed consent was obtained from all study participants. Women with underlying diseases influencing vitamin D metabolism (bowel disease, liver disease, chronic kidney disease, diabetes mellitus types I and II, chronic hypertension, systemic lupus erythematodes, conditions causing hyperparathyroidism), any vitamin D substitution over 800 IU/day, a history of surgery involving the thyroid gland, and pregnancies affected by chromosomal abnormalities or fetal anomalies were excluded. Maternal demographic characteristics were assessed at study inclusion. Study participants were recruited at their booking visit at the Department of Obstetrics and Fetomaternal Medicine at the Medical University of Vienna between 11 + 0 and 13 + 6 weeks of gestation. All pregnancies were dated according to the crown-to-rump length measured during the first trimester screening performed by certified sonographers. Patients underwent 4 prenatal and 1 postpartum study visits. Assessments at prenatal visits were conducted at 11 + 0 to 13 + 6, 18 + 0 and 22 + 6, and 28 + 0 to 32 + 0 weeks of gestation and at delivery, including evaluation of maternal baseline characteristics (height, weight, BMI), BP, urine protein excretion, and blood sampling of 25-hydroxyvitamin D, as well as a food frequency questionnaire (FFQ). Serum 25-hydroxyvitamin D was measured using the chemiluminescence immunoassay (CLIA) DiaSorin on the Liaison XL analyzer. All patients had an anomaly scan between 18 + 0 and 22 + 6 weeks of gestation, and a routine fetal biometry was performed at 28 + 0 to 32 + 0 gestational weeks. The estimated fetal weight (EFW) was calculated using the Hadlock formula [16]. The postpartum assessment was conducted at a follow-up visit 12 weeks postpartum and included a final evaluation of the aforementioned parameters. The primary outcome of the study was the evaluation of 25(OH)D levels over the course of pregnancy and in the postpartum period. Secondary outcomes included the evaluation of the intake of prenatal vitamins containing vitamin D, the evaluation of dietary vitamin D intake, and the association of vitamin D deficiency with complications during pregnancy, such as gestational diabetes mellitus (GDM), preterm birth, preterm labor, preterm rupture of membranes (PROM), hypertensive disorders of pregnancy (HDP), and small for gestational age (SGA), as well as neonatal complications (respiratory distress, admission to the neonatal intensive care unit). Furthermore, we assessed the association of vitamin D deficiency with the following parameters: birth weight, EFW (2nd and 3rd trimesters). Outcomes were assessed for singleton and twin pregnancies separately. The vitamin D status was defined as follows: values above 75 nmol/L were classified as sufficient vitamin D status, between 75 and 50 nmol/L as insufficient vitamin D status, and below 50 nmol/L as vitamin D deficient [19]. HDP were diagnosed according to the guidelines published by the International Society for Hypertensive Disorders of Pregnancy (ISSHP) [20]. Birth weight centiles were calculated according to centiles published by Poon et al. [21]. SGA was defined as a birth weight <10th centile after correcting for gestational age [22]. Information about delivery and perinatal and maternal outcomes was obtained using the obstetrical documentation system Viewpoint (Viewpoint 5.6.8.428, Wessling, Germany) and neonatal records.

### Statistical Analysis

Continuous variables were described using the median and IQR. Group comparisons were performed using the Mann–Whitney *U* test. Categorical variables were described as absolute and relative frequencies, and the chi-square test was performed for group comparison. For the evaluation of mean 25(OH)D levels over the course of pregnancy, we calculated the pairwise difference between all study visits during pregnancy. For these differences, we analyzed simultaneous 95% confidence intervals (CI) and adjusted *p*-values using a generalization of Tukey’s method [23,24]. To account for dependences of repeated measurements within study participants, the required standard errors and covariates were calculated using robust variance estimation. The same method was also applied in a logistic regression model to analyze the differences in the percentage of women with vitamin D sufficiency, insufficiency, and deficiency in between study visits. To evaluate an association of vitamin D deficiency with complications of pregnancy, we performed a correlation of first-trimester 25(OH)D deficiency and complications during pregnancy using cross tables and chi-square tests. For all calculations, the statistical software IBM SPSS version 25 (IBM Corporation, Armonk, NY, USA) and R version 3.5 (R Foundation for Statistical Computing, Vienna, Austria) were used.

## 3. Results

### 3.1. Patient Characteristics and Pregnancy Outcomes

A total of 251 healthy pregnant women were recruited, 199 singleton pregnancies and 52 twin pregnancies. One singleton and one twin pregnancy were excluded due to fetal anomalies, leaving 198 singleton and 51 twin pregnancies (n = 249). A total of 967 study visits were conducted. The subcohort of twin pregnancies consisted of 32 (62.8%) dichorionic diamniotic, 17 (33.3%) monochorionic diamniotic, and 2 monochorionic monoamniotic (3.9%) twin pregnancies. Baseline patient characteristics are summarized in Table 1. Pregnancy outcomes were analyzed separately for singleton and twin pregnancies (see Appendix A).

### 3.2. Vitamin D Deficiency during Pregnancy

An analysis of all study visits showed median 25(OH)D levels of 43.6 nmol/L (IQR 27.6–59.7 nmol/L). Overall, 59.5% (n = 147) of the whole study population evaluated over the course of pregnancy and postpartum demonstrated 25(OH)D levels that fulfil the criteria for vitamin D deficiency (as seen in Figure 1). The vitamin D status (defined as sufficient vitamin D status, insufficient vitamin D status and, deficient vitamin D status, as previously described) was analyzed for changes over the course of pregnancy by comparing study visits. The majority of all study participants showed vitamin D deficiency at each study visit, ranging from 59.5% (n = 147) at visit 1 to up to 66.9% (n = 103) at visit 4. The analysis of all 25(OH)D levels measured from visits 1 to 5 in twin pregnancies revealed median levels of 47.6 nmol/L (IQR 26.43–62.05 nmol/L) in twin pregnancies compared to a median of 43.0 nmol/L (IQR 27.6–58.9) in singletons. There was no significant difference in 25(OH)D levels over the course of pregnancy and postpartum between singleton and twin pregnancies (*p* = 0.09).

Our analysis revealed that women with vitamin D deficiency showed a significantly higher BMI at booking (median 23.99 kg/m^2^, IQR 21.09–27.64 vs. median 21.48 kg/m^2^, IQR 20.2–24.84; *p* < 0.001). Vitamin D deficiency was, furthermore, significantly more common in smokers (80%) than in non-smokers (55.5%) (*p* < 0.001), and spontaneous conception was significantly (*p* < 0.001) more often associated with vitamin D deficiency (62.3%) than assisted reproduction (47.4%). Nulliparous pregnant women showed significantly (*p* < 0.001) lower rates of vitamin D deficiency (54.6%) than multiparous pregnant women (64.8%). Ethnicity did not have a significant influence (*p* = 0.21). Within our cohort, vitamin D deficiency was not significantly associated with biometric changes in first-trimester (singleton *p* = 0.906; twin fetus 1 *p* = 0.327, twin fetus 2 *p* = 0.24), second-trimester (singleton EFW *p* = 0.106; twin fetus 1 EFW *p* = 0.44, twin fetus 2 EFW *p* = 0.46), and third-trimester (singleton EFW *p* = 0.38; twin fetus 1 EFW *p* = 0.14, twin fetus 2 EFW *p* = 0.63) ultrasound in twin and singleton pregnancies.

The assessment of factors influencing the vitamin D status, such as exposure to sunlight reflected by a suntan in the past 3 months, skin coverage with clothing, usage of sunscreen, and season of blood sampling, revealed the following: Covered women showed significantly (*p* = 0.003) lower median 25(OH)D levels (median 31.65 nmol/L; IQR 15.9–46.2) than uncovered women (median 45.9 nmol/L; IQR 29.1–61.6). Women reporting a suntan in the past 3 months revealed significantly (*p* < 0.001) higher 25(OH)D levels (median 59.0 nmol/L; IQR 45.33–70.48) than women without a suntan (median 39.05 nmol/L; IQR 25.08- 52.83). Patients reporting regular usage of sunscreen showed significantly (*p* < 0.001) higher levels of 25(OH)D (median 51.0 nmol/L; IQR 36.75–62.0) compared to women who denied any use of sunscreen (median 37.0 nmol/L; IQR 22.2–51.6). Seasonal influence was ruled out as blood sampling was equally distributed between seasons (spring 26%, summer 25%, autumn 26%, winter 23%), with no significant difference between the time of study visits 1 to 5 (*p* = 0.09). Subgroup analysis for twin pregnancies did not reveal significant differences between singleton and twin pregnancies concerning aforementioned factors influencing the vitamin D status.

The investigation of a possible association of vitamin D deficiency with maternal pregnancy complications, such as preterm labor, preterm birth, PROM, HDP, and GDM, as well as perinatal adverse outcomes, such as low birth weight (SGA), NICU admission, and RDS, showed no significant association of first-trimester vitamin D deficiency with maternal or neonatal adverse outcomes both in singleton and in twin pregnancies (as seen in Table 2 and Appendix A).

### 3.3. Vitamin D Supplementation and Nutritional Vitamin D Intake

Vitamin D intake during pregnancy was assessed via a food frequency questionnaire (FFQ), and the daily vitamin D intake was calculated. A total of 241 questionnaires were evaluated. Overall, 160 study participants (66.4% of the total study population) reported intake of pregnancy vitamins containing vitamin D. Daily vitamin D supplementation reflected a consumption of a median of 10 micrograms of vitamin D per day (IQR 5–10 micrograms). Nutritional vitamin D intake was calculated via the FFQ in the first and second trimesters, as well as at the time of delivery. Our analysis revealed low nutritional vitamin D intake with a median of 2.3 micrograms (IQR 1.4–3.13 micrograms) in the first trimester, 2.15 micrograms (IQR 1.33–3.40 micrograms) in the second trimester, and 2.4 micrograms (IQR 1.4–6.1 micrograms) at the time of delivery. The 25(OH)D levels were significantly (*p* < 0.001) higher in women who reported intake of pregnancy vitamins containing vitamin D (median 51.05 nmol/L, IQR 36.7–63.9) compared to women who did not use pregnancy vitamins (median 37.0 nmol/L, IQR 23.4–56.2) but still did not reach sufficient vitamin D levels. Subgroup analysis for vitamin D supplementation in twin pregnancies compared to singleton pregnancies demonstrated no significant differences in the rates of vitamin D supplementation (*p* = 0.45) or the amount of supplemented vitamin D (*p* = 0.68). Multivariable linear regression analysis was performed to identify and jointly analyze the factors influencing 25(OH) D levels during pregnancy, as shown in Table 3. The coefficients are estimates of the mean differences due to each influencing factor when the values of the other influencing factors are constant. The supplementation of pregnancy vitamins containing vitamin D could be related to a 11.7 nmol/L increase in 25(OH) D levels. Similarly, sunlight exposure in the summer raised 25(OH)D levels by 9.9 nmol/L, and a suntan in the past 3 months went along with a rise of 9.2 nmol/L in 25(OH) D levels. On the contrary, the season of winter, covering the skin, and a higher BMI at booking lowered 25(OH) D levels by 4.4, 10.2, and 0.7 nmol/L, respectively.

## 4. Discussion

Vitamin D deficiency has been widely reported during pregnancy and has been identified as an area of concern as it poses a possible risk factor for adverse maternal and perinatal outcomes, including preeclampsia, fetal growth restriction, and preterm birth [2,14]. Vitamin D supplementation is a safe intervention during pregnancy [25]. In this prospective cohort study, we were able to generate longitudinal data on serum vitamin D levels during pregnancy and postpartum in low-risk pregnant women at the Medical University of Vienna. To the best of our knowledge, this is the first prospective cohort study evaluating longitudinal 25(OH)D levels during pregnancy and postpartum in Austria, using a standardized reference range of below 50 nmol/L to define vitamin D deficiency, also assessing the nutritional and environmental determinants of vitamin D status in pregnant women and postpartum.

In this analysis, 59.5% of pregnant women were classified as vitamin D deficient in the first trimester, 54.8% in the second trimester, 58.5% in the third trimester, 66.9% at birth, and 60% 12 weeks postpartum. Our data are comparable to findings by Cabaset et al., describing 63.5% of vitamin D deficiency in pregnant women in the first trimester and 54.4% in the third trimester in Switzerland [26,27]. Furthermore, the rates of vitamin D deficiency within our cohort were found to be proportionate to other European countries, such as Germany, reporting a prevalence of 77% [28]; Northern Italy with 85% [29]; and France, which reported 41% [30] of vitamin D deficiency during pregnancy. Furthermore, we were able to show that twin pregnancies are not at higher risk of vitamin D deficiency compared to singleton pregnancies.

We showed a high prevalence of vitamin D deficiency within our cohort, despite vitamin D supplementation in the majority of study participants. Within our study cohort, we were able to show a significant impact of vitamin D supplementation on 25(OH)D levels, in line with a recent meta-analysis showing evidence that maternal vitamin D supplementation significantly increases maternal and infant concentrations [31]. However, dietary vitamin D intake as well as vitamin D supplementation failed to achieve sufficient vitamin D uptake within our cohort. Moreover, our findings confirm that pregnant women are unable to reach sufficient vitamin D levels through sunlight exposure and nutritional vitamin D intake alone, supporting the need for adequate individualized supplementation. We provided evidence that vitamin D supplementation in pregnant women in Austria, with recommended supplementation of 200–800 IU per day [17,18], is insufficient and needs further attention. A study assessing the compliance of patients in vitamin supplementation during pregnancy also found that recommendations regarding supplementation during pregnancy and the periconceptional period are still insufficient [32]. The study group of Mansur et al. recently recommended reaching a level of 30 ng/mL or more throughout pregnancy, even suggesting target levels >40 ng/mL [3]. Various studies assessing the safety of vitamin D supplementation in pregnancy, have confirmed supplementation of 25(OH)D up to 4400 IU/day to be safe [33,34]. There are ongoing trials that, when published, will hopefully provide knowledge to support consistent recommendations for regimens of vitamin D supplementation for women during pregnancy [35].

There has been evidence of an association of vitamin D deficiency with different pregnancy-related disorders, such as hypertensive disorders of pregnancy, gestational diabetes, fetal growth restriction, and preterm birth, as well as common complications of the neonatal period [8,12,36,37,38]. Kinshella et al. recently presented evidence that maternal vitamin D deficiency poses a nutritional risk factor for preeclampsia. Vitamin D supplementation and healthy maternal diet were identified as protective nutritional factors associated with a lower risk of developing preeclampsia [37]. In line with these findings, the FEPED Study, assessing the effects of the maternal vitamin D status in the first trimester of pregnancy on the risks of preeclampsia, GDM, preterm birth, and SGA at birth, found a significantly decreased risk of preeclampsia associated with maternal serum 25(OH)D levels of ≥30 ng/mL in the first and third trimesters, while no significant linear relationship was identified for GDM, preterm birth, and SGA [39]. A randomized trial by Rostami et al. was able to show a significant decrease in adverse pregnancy outcomes including GDM, HDP and preterm birth in patients with higher vitamin D supplementation compared to vitamin D deficient pregnant women [40]. Interestingly, the high rates of vitamin D deficiency observed in our cohort did not lead to higher rates of complications during pregnancy. The 25(OH)D levels were not correlated with an increased risk of maternal or neonatal pregnancy complications and did not lead to a higher incidence of adverse perinatal outcomes. Our results are in accordance with a study by Nassar et al., who found no association of vitamin D deficiency in the first trimester with complications of pregnancy [41]. A recent study by Lee et al. revealed that maternal vitamin D deficiency was associated with lower birth weight and smaller head circumference [42]. We found no significant impact of vitamin D deficiency on biometric measurements or fetal growth and birth weight within our cohort. These results are in line with data by Roth et al., stating that there is no robust evidence for vitamin D showing effects on low birth weight, length, and head circumference at birth [43]. However, our study was not powered for the assessment of the effect of vitamin D deficiency on pregnancy-related complications. We believe that the strengths of our investigations were the longitudinal character of the study, including the assessment of first, second, and third trimesters, as well as the postpartum assessment of vitamin D levels within a reasonably large cohort of healthy pregnant women. Furthermore, to the best of our knowledge, this is the first study incorporating the assessment of nutritional vitamin D intake over the course of pregnancy and postpartum.

## 5. Conclusions

A high prevalence of vitamin D deficiency among pregnant women and its possible effects on pregnancy outcomes and offspring health have been identified as an area of primary concern. In this longitudinal observational cohort study evaluating 25(OH)D levels in healthy pregnant women in Austria, we showed that healthy pregnant women suffer from a concerningly high rate of vitamin D deficiency despite vitamin D supplementation in the majority. Recommended amounts of vitamin D supplementation are not covered with common supplementary products and are not sufficient to ensure adequate vitamin D levels in pregnant women in Austria. Furthermore, dietary vitamin D intake during pregnancy was low and did not significantly impact the serum 25(OH)D levels within our cohort, further supporting the need for adequate vitamin D supplementation. Hence, individualized vitamin D assessment during pregnancy should be considered to ensure adequate supplementation and prevention of vitamin D deficiency. Moreover, vitamin D supplementation during pregnancy should be encouraged by clinicians, with a special focus on pregnant women at high risk of vitamin D deficiency, such as smokers, women with a high BMI, and women wearing extensive covering.

## Figures and Tables

**Figure 1 nutrients-15-00339-f001:**
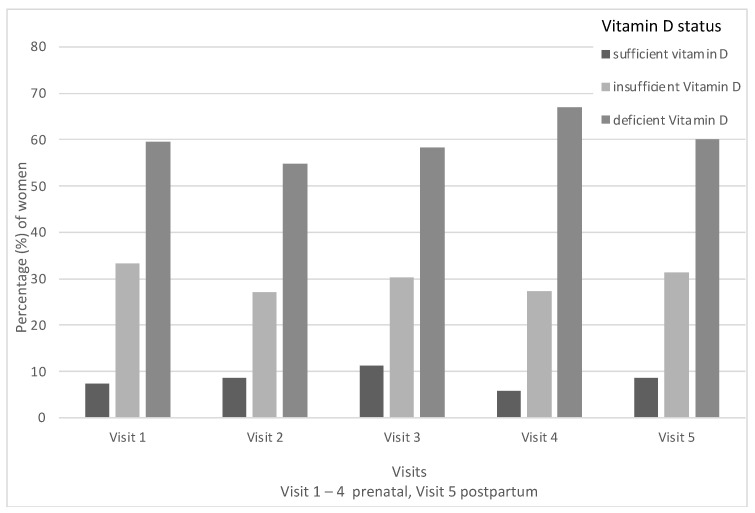
Vitamin D status over the course of pregnancy and postpartum.

**Table 1 nutrients-15-00339-t001:** Patient characteristics (singleton and twin pregnancies).

	Study Participants n = 249
GA at study inclusion in weeks, median (IQR)	12.43 (12.0–13.0)
Maternal age in years, median (IQR)	31 (26–34)
BMI in kg/m^2^, median (IQR), booking	23.31 (20.55–27.05)
Ethnicity	
Caucasian, n (%)	221 (94)
African, n (%)	3 (1.3)
Asian, n (%)	8 (3.4)
Hispanic, n (%)	2 (0.9)
Mixed, n (%)	1 (0.4)
Conception	
Spontaneous, n (%)	208 (87.8)
Assisted, n (%)	29 (12.2)
Smoking, n (%)	40 (16.1)
Nulliparous, n (%)	113 (47.7)
Systolic blood pressure, mmHg, median (IQR)	120 (112–127)
Diastolic blood pressure, mmHg, median (IQR)	78 (72–84)
Twin pregnancy, n (%)	51 (20.5)
Monochorionic diamniotic, n (%)	17 (33.3)
Monochorionic monoamniotic, n (%)	2 (3.9)
Dichorionic diamniotic, n (%)	32 (62.8)

GA (gestational age), IQR (interquartile range), BMI (body mass index).

**Table 2 nutrients-15-00339-t002:** Vitamin D deficiency in the first trimester and association with complications in singleton pregnancies presented as absolute frequencies and percentages in parentheses.

	Vitamin D < 50 nmol/L n = 114	Vitamin D > 50 nmol/L n = 67	*p*-Value
Preterm birth, n (%)	12 (10.5)	5 (7.5)	0.49
HDP, n (%)	6 (5.3)	2 (3.0)	0.47
GDM, n (%)	13 (11.4)	4 (6.0)	0.23
IGDM, n (%)	12 (10.5)	3 (4.5)	0.15
PROM, n (%)	10 (8.8)	8 (11.9)	0.49
Preterm labor, n (%)	5 (4.4)	4 (6.0)	0.64
Cervical insufficiency, n (%)	2 (1.8)	1 (1.5)	0.89
NICU admission, n (%)	6 (5.3)	2 (3.0)	0.46
RDS, n (%) SGA, n (%)	6 (5.3) 17 (15.2)	3 (4.5) 6 (9.2)	0.80 0.26

HDP (hypertensive disorders of pregnancy), GDM (gestational diabetes mellitus), IGDM (insulin-dependent gestational diabetes mellitus), PROM (preterm rupture of membranes), NICU (neonatal intensive care unit), RDS (respiratory distress syndrome), SGA (small for gestational age).

**Table 3 nutrients-15-00339-t003:** Multivariable linear regression analysis for factors influencing vitamin D levels during pregnancy.

	Mean Change nmol/L	95% CI	*p*-Value
Visit 2	0.69	−1.56–2.93	0.55
Visit 3	0.31	−2.47–3.08	0.83
Visit 4	−2.89	−6.26–0.48	0.09
Vitamin D supplementation	11.72	7.92–15.51	<0.001
Summer	9.91	6.0–13.81	<0.001
Winter	−4.4	−7.73–1.07	0.009
Autumn	3.64	−0.25–7.52	0.07
Suntan in the past 3 months	9.16	4.62–13.70	<0.001
BMI at booking	−0.68	−1.07–0.30	<0.001
Covering	−10.19	−17.28–3.11	0.005

## Data Availability

Data are available upon request.

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
