# Peer review of "Longitudinal Assessment of Serum 25-Hydroxyvitamin D Levels during Pregnancy and Postpartum—Are the Current Recommendations for Supplementation Sufficient?"

_nutrients, 2023, doi:10.3390/nu15020339_

Round 1

Reviewer 1 Report

Thank you very much for giving me the opportunity to read this article, first of all congratulations for the work done.

Secondly, some aspects to improve, I would like to start with the references, I think there is a lot of margin of years, there are some from 2000 and the most current ones from 2020, in my opinion they should be less than 10 years old, could the authors update the references?

Introduction is correct

There were no drop-outs, i.e. no women who did not want to continue in the study.

In the Material and Methods section it is stated that the study was approved by the local research ethics committee (approval number 1051/2011) of the Medical University of Vienna, but nowhere is it stated whether the women were given informed consent.

I think the results are adequate as well as the discussion.

Reviewer 2 Report

The study deals with the currently frequently discussed issue of saturation of the body of pregnant women with vitamin D.

Remarks:

- in the Introduction, the 25(OH)D concentration values given in nmol/L are different than in ng/ml (line 37). This should be unified.

- the sentence "...with serum 25(OH)D targets of > 25-50 nmol/L (10-20 ng/ml)" is incomprehensible (lines 60-61). If the target concentration would already be above 25 nmol/L, why did the authors assume that the vitamin D deficit is below 50 nmol/L?

- it should be added whether there are any recommendations regarding vitamin D supplementation by pregnant women in Austria, and if so, what are the recommendations, and in the discussion of the results it should be compared to these recommendations

- in Materials and Methods, vitamin D units (800 IE/day) should be corrected (line 76)

- there is no information on the method of assessing the concentration of vitamin D! The methodology used in this regard should be specified in detail.

- the vertical axis of Figure 1 is not well readable (technical error), but it seems that it would be better to give the percentage of women, not the number of women. On the horizontal axis, it would also be worth noting that visits 1-4 concern pregnant women, and visit 5 women after childbirth (this would be clearer). In addition, the spelling of vitamin D (capital letter) should be corrected in the legend.

- figure No. 2 referred to in the text is missing in the Results ??

- how the authors explain the fact that "Patients reporting regular usage of sunscreen showed significantly (p<0.001) higher levels of 25(OH)D (median 51.0 nmol/L) compared to women that denied any use of sunscreen (median 37.0 nmpl/L)” (lines 178-180).

- whether there were women who took single vitamin D preparations (such preparations usually have a higher dose of this vitamin) or only vitamins containing vitamin D , and what percentage of women used vitamin D supplementation postpartum.

- why the authors analyzed only the effect of vitamin D concentration in the first trimester on the course of pregnancy, and not in all trimesters?

- what BMI is meant in the sentence "... the covering of skin and BMI lowered 25(OH) D" (line 221)

- In Discussion, comparing the results to a study that used a concentration below 15 nmol/L  as a vitamin D deficiency in my opinion is currently highly debatable. Such a study can only be quoted to illustrate how the approach to vitamin D levels in science is changing

Reviewer 3 Report

The manuscript „Longitudinal assessement of serum 25-Hydroxyvitamin D levels in pregnancy and postpartum – are the current recommendations for supplementation sufficent?” evaluate maternal 25-(OH)D levels, effects of dietary and supplementary vitamin D intake over the course of pregnancy.

My comments/suggestions

1.     Different units for vitamin D concentrations are present in manuscript file (IU, ng/mL, nmol/L). Could you give the information about „IE/day”, please? (line 76).

2.     In The Aim of manuscript Authors have written „we investigated a possible association of vitamin D deficiency with maternal and perinatal complications”. Could you show the fragment in manuscript where this is explained, please? Or add it in the text, please.

3.     One of the important information is the season of year, because vitamin D can have different concentrations. Do the authors have such information? Only Table 3 presents any seasonal data.

4.     It is worth to add in Conclussions the limitations of research.
